# Discordance between hyposalivation and xerostomia among community-dwelling older adults in Japan

Ichizo Morita[1]*, Hisayoshi Morioka[2], Yoshikazu Abe[3], Taketsugu Nomura[3], Seiji Nakashima[3], Iwane Sugiura[3], Yujo Inagawa[3], Yuka Kondo[4], Chisato Kameyama[5], Kanae Kondo[1], Naoji Kobayashi[1]

1 Japanese Red Cross Toyota College of Nursing, Toyota, Aichi, Japan, 2 Department of Public Health, Institute of Health Biosciences, The University of Tokushima Graduate School, Tokushima, Tokushima, Japan, 3 Gifu Prefecture Dental Association, Gifu, Gifu, Japan, 4 Gifu Prefecture Medical Association, Gifu, Gifu, Japan, 5 Gifu Pharmaceutical Association, Gifu, Gifu, Japan

* i-morita@rctoyota.ac.jp

**Data Availability Statement:** All data are contained within the paper and its Supporting Information files.

## Abstract

Individuals with an objective decrease in salivary flow (objective dry mouth) may not be aware of subjective dry mouth (xerostomia). However, no clear evidence exists to explain the discordance between subjective and objective dry mouth. Therefore, this cross-sectional study aimed to assess the prevalence of xerostomia and decreased salivary flow among community-dwelling elderly adults. In addition, this study assessed several potential demographic and health status determinants of the discrepancy between xerostomia and reduced salivary flow. The 215 participants in this study were community-dwelling older people aged 70 years and above who underwent dental health examinations between January-February 2019. Symptoms of xerostomia were collected in the form of a questionnaire. The unstimulated salivary flow rate (USFR) was measured by a dentist using visual inspection. The stimulated salivary flow rate (SSFR) was measured using the Saxon test. We identified 19.1% of participants as having mild-severe USFR decline with xerostomia and 19.1% as having mild-severe USFR decline without xerostomia. Additionally, 26.0% of participants had low SSFR and xerostomia, and 40.0% had low SSFR without xerostomia. Except for the age trend, no factors could be associated with the discordance between USFR measurement and xerostomia. Furthermore, no significant factors were associated with the discordance between the SSFR and xerostomia. However, females were significantly associated (OR = 2.608, 95% CI = 1.174–5.791) with low SSFR and xerostomia, as compared to males. Age was a factor that was also significantly associated (OR = 1.105, 95% CI = 1.010–1.209) with low SSFR and xerostomia. Our findings indicate that approximately 20% of the participants had low USFR without xerostomia, and 40% had low SSFR without xerostomia. This study showed that age, sex, and the number of medications may not be factors in the discrepancy between the subjective feeling of dry mouth and reduced salivary flow.

**Funding:** The data for this study were obtained from a survey conducted by the Public Interest Incorporated Foundation 8020 Promotion Foundation Dental Health Activity Project in 2018 (No.9, YA). This work was supported by the Japanese Red Cross Toyota College of Nursing Fund for Research Support, and the funding was received by I M. The funders had no role in study design, data collection and analysis, decision to publish, or preparation of the manuscript.

**Competing interests:** The authors have declared that no competing interests exist.

## Introduction

Current classifications of dry mouth include xerostomia, the subjective symptom of dry mouth, measurable only by interviewing individuals [1,2], and salivary gland hypofunction, an objective diagnosis determined by measuring salivary flow rate [3]. These two manifestations of dry mouth are not necessarily concurrent [4]. Indeed, epidemiological studies have reported that the proportion of those with both symptoms is extremely low (11–15%) [5,6]. In addition, objective dry mouth is measured by adding together the unstimulated salivary flow rate (USFR) and the stimulated salivary flow rate (SSFR) [1,7,8]. This branching of the definition of dry mouth further complicates our understanding of the notion of dry mouth.

Based on the presence or absence of xerostomia and objective dry mouth, there are four categories of individuals. First, patients that do not present xerostomia or objective reduction in salivation do not need medical attention for dry mouth. Secondly, patients with both xerostomia and objective decrease in salivary flow experience consistent symptoms and will undoubtedly need medical attention for dry mouth. Thirdly, patients with xerostomia lacking an objective reduction in salivation will also receive medical treatment because they are aware of this problem.

Lastly, patients who do not experience xerostomia despite an objective decrease in salivary flow may not experience dry mouth symptoms, and in many cases, they may not seek medical care. Identifying these patients may help prevent the symptoms and illnesses that result from xerostomia [4,9,10] and help recognize dehydrated older people [11], as well as patients with diabetes, thyroid disease [12], and renal disease [13]. Symptoms caused by decreased salivary flow (e.g., oral pain, burning and stinging sensations, difficulty swallowing and speaking, and taste alterations) result in reduced quality of life [1]. Currently, there is no clear evidence to explain the discordance between xerostomia and objective dry mouth.

This study aimed to assess the prevalence of xerostomia and reduced salivary flow among community-dwelling elderly in Japan. Moreover, the study evaluated several potential demographic and health status determinants of the discrepancy between xerostomia and reduced salivary flow.

## Materials and methods

### Study participants

The study included 215 community-dwelling individuals (98 males, 117 females; age range 72–94 years, mean age 80.2±4.5 years) who visited one of the ten dental clinics included in the study from the Chubu region of Japan. The inclusion criteria were as follows: community-dwelling, independent, age 70 years or older and provided informed consent to participate in the study. The exclusion criteria were as follows: hearing or language impairment that could affect the interview, physical or mental disability that could interfere with the oral exam, symptoms, or need for treatment for dental disease. The participants underwent dental health examinations in the period January-February 2019. Informed written consent was obtained from all participants.

### Dry mouth measurements

The presence of xerostomia was assessed with one question: "Do you worry about having a dry mouth?" [14–16]. Answer options were either yes or no.

The objective salivary flow rate measurements were conducted using two tests: the unstimulated salivary flow rate (USFR) and the stimulated salivary flow rate (SSFR). USFR measurement was performed by a dentist by visual inspection of the oral cavity. This method uses

a modified clinical diagnosis classification scale for tongue mucosa conditions [17]. The three evaluation stages were divided into two groups: a normal group and a mild-severe decline group.

SSFR measurements were performed using the Saxon test [18]. The participants chewed sterile gauze (7.5 x 7.5 cm, 12-Ply) for 2 min, and the amount of secreted saliva was calculated from the change in the weight of the gauze before and after chewing. Saliva weights less than or equal to 2 g/2 min were considered low salivary flow, while more than 2 g saliva/2 min was considered normal salivary flow.

### Health status assessment

Information on the number of medications taken was obtained from the participants' prescription recording notebooks. Patients were divided into two groups: less than five medications and five or more medications. The number of diseases under treatment was self-reported from a list of 26 diseases and descriptive fields. The participants were divided into three groups: no disease under treatment, one or two, and three or more.

These questions and measures were included in the oral health survey of community-dwelling elderly, conducted by the local government and dental association. The present study was a secondary analysis of the data and was approved by the Ethics Committee of the Japanese Red Cross Toyota College of Nursing (project no. 1910). This study was conducted April-September 2022. The authors did not have access to any information that could identify individual participants during or after data collection.

### Statistical analysis

Participants were classified into four groups based on subjective and/or objective dry mouth determination, which were further classified into two categories: normal and decreased. Multinomial logistic regression analysis was performed, with the four groups of dry mouth as the dependent variables and age, sex, and a number of medications as explanatory variables. The analysis was performed separately for the USFR and SSFR. Analyses were performed using SPSS Statistics 28 (IBM, Armonk, New York, USA) and R version 4.0.2. (The R Foundation for Statistical Computing).

## Results

### Characteristics of the study participants

Of the total number of participants, approximately 50% were between 70–79 years old, while the other half were over 80 years old (Table 1). There were 111 (51.6%) participants taking less than five medications and 104 (48.4%) participants taking five or more medications. Furthermore, 23 (10.7%) had no disease under treatment, 133 (61.9%) had one or two diseases under treatment, and 59 (27.4%) were treated for three or more diseases.

A high correlation coefficient was found between the number of medications and the number of diseases under treatment (Spearman's rank correlation coefficient, $\rho = 0.582$, $p < 0.001$). Therefore, to avoid multicollinearity, the number of diseases under treatment was excluded from the explanatory variables of multinomial logistic regression analysis.

### Proportion of hyposalivation and xerostomia

First, we classified 19.1% (95% confidence interval (CI) = 14.0–25.0) of participants into the mild-severe USFR decline group with xerostomia (Table 2). Secondly, the rate of participants

**Table 1. Characteristics of the study participants.**

| Variables | | Mean | Standard deviation |
|---|---|---|---|
| **Age (years)** | | **80.2** | **4.5** |
| | | **Number** | **Percentage (%)** |
| Age categories (years) | 70–79 | 113 | 52.6 |
| | 80–89 | 93 | 43.3 |
| | ≥90 | 9 | 4.2 |
| Sex | Males | 98 | 45.6 |
| | Females | 117 | 54.4 |
| Number of medications | <5 medications | 111 | 51.6 |
| | ≥5 medications | 104 | 48.4 |
| Number of diseases under treatment | None | 23 | 10.7 |
| | 1–2 | 133 | 61.9 |
| | ≥3 | 59 | 27.4 |

Total n = 215.

with mild-severe USFR decline without xerostomia was 19.1%. Lastly, the rate of participants with normal USFR and xerostomia was 17.7% (95% CI = 12.8–23.4).

The proportion of participants with low SSFR and xerostomia was 26.0% (95% CI = 20.3–32.5), while the proportion of participants with low SSFR without xerostomia was 40.0% (95% CI = 33.4–46.9), the highest proportion of the four classifications. The proportion of participants with normal SSFR and xerostomia was 10.7% (95% CI = 6.9–15.6).

## Factors associated with the discordance between objective salivary flow rate measurement and subjective symptoms of xerostomia

There were no significant factors associated with the discordance between the USFR and xerostomia (Table 3), although we did identify an age trend among these participants (odds ratio

**Table 2. Rate of unstimulated and stimulated salivary flow rate by symptoms of xerostomia.**

| | | | | Subjective symptoms of xerostomia | | |
|---|---|---|---|---|---|---|
| | | | | Do you worry about having a dry mouth? | | |
| **Objective salivary flow rate measurement** | | | | **Yes** | **No** | **Total** |
| | Unstimulated salivary flow rate (USFR) | Mild-Severe decline | n | 41 | 41 | 82 |
| | | | % | 19.1 | 19.1 | 38.1 |
| | Inspection by a dentist | | 95% CI | 14.0–25.0 | 14.0–25.0 | |
| | | Normal | n | 38 | 95 | 133 |
| | | | % | 17.7 | 44.2 | 61.9 |
| | | | 95% CI | 12.8–23.4 | 37.4–51.1 | |
| | Stimulated salivary flow rate (SSFR) | ≤2g/2 min | n | 56 | 86 | 142 |
| | | | % | 26.0 | 40.0 | 66.0 |
| | Saxon test | | 95% CI | 20.3–32.5 | 33.4–46.9 | |
| | | >2g/2 min | n | 23 | 50 | 73 |
| | | | % | 10.7 | 23.3 | 34.0 |
| | | | 95% CI | 6.9–15.6 | 17.8–29.5 | |
| | Total | | n | 79 | 136 | |
| | | | % | 36.7 | 63.3 | |

CI: Confidence interval.

**Table 3. Factors associated with the discordance between unstimulated salivary flow rate and subjective symptoms of xerostomia.**

| Unstimulated salivary flow rate (USFR) | Subjective symptoms of xerostomia | Factors | | OR | 95% CI | P-value |
|---|---|---|---|---|---|---|
| Low | Yes | Age | Year | 1.056 | 0.972–1.147 | 0.198 |
| | | Sex | Males | Ref | | |
| | | | Females | 2.085 | 0.970–4.480 | 0.060 |
| | | Number of medications | <5 medications | Ref | | |
| | | | ≥5 medications | 0.728 | 0.341–1.555 | 0.412 |
| Normal | Yes | Age | Year | 1.084 | 0.998–1.177 | 0.057 |
| | | Sex | Males | Ref | | |
| | | | Females | 1.339 | 0.622–2.883 | 0.455 |
| | | Number of medications | <5 medications | Ref | | |
| | | | ≥5 medications | 1.517 | 0.702–3.276 | 0.289 |
| Low | No | Age | Year | 1.013 | 0.930–1.104 | 0.764 |
| | | Sex | Males | Ref | | |
| | | | Females | 1.576 | 0.750–3.313 | 0.230 |
| | | Number of medications | <5 medications | Ref | | |
| | | | ≥5 medications | 1.349 | 0.644–2.826 | 0.428 |

Reference category for dependent variable: Group with normal unstimulated salivary flow rate (USFR) and without subjective symptoms of xerostomia.

OR: Odds ratio, CI: Confidence interval, Ref: Reference.

(OR) = 1.084, 95%CI = 0.998–1.177, per 1-year increase). Except for this age trend, no other factors could be associated with the discordance between USFR measurements and xerostomia. Interestingly, females tended to have a lower USFR and xerostomia (OR = 2.085, 95% CI = 0.970–4.480).

No significant factors were associated with the discordance between SSFR and xerostomia (Table 4). However, females were significantly associated (OR = 2.608, 95% CI = 1.174–5.791)

**Table 4. Factors associated with the discordance between stimulated salivary flow rate and subjective symptoms of xerostomia.**

| Stimulated salivary flow rate (SSFR) | Subjective symptoms of xerostomia | Factors | | OR | 95% CI | P-value |
|---|---|---|---|---|---|---|
| Low | Yes | Age | Year | 1.105 | 1.010–1.209 | 0.029 |
| | | Sex | Males | Ref | | |
| | | | Females | 2.608 | 1.174–5.791 | 0.019 |
| | | Number of medications | <5 medications | Ref | | |
| | | | ≥5 medications | 0.885 | 0.403–1.945 | 0.761 |
| Normal | Yes | Age | Year | 1.036 | 0.922–1.165 | 0.549 |
| | | Sex | Males | Ref | | |
| | | | Females | 1.254 | 0.464–3.388 | 0.655 |
| | | Number of medications | <5 medications | Ref | | |
| | | | ≥5 medications | 0.902 | 0.334–2.434 | 0.838 |
| Low | No | Age | Year | 1.030 | 0.946–1.121 | 0.498 |
| | | Sex | Males | Ref | | |
| | | | Females | 1.731 | 0.854–3.505 | 0.128 |
| | | Number of medications | <5 medications | Ref | | |
| | | | ≥5 medications | 0.909 | 0.449–1.839 | 0.790 |

Reference category for dependent variable: Group with normal stimulated salivary flow rate (SSFR) and without subjective symptoms of xerostomia.

OR: Odds ratio, CI: Confidence interval, Ref: Reference.

with low SSFR and xerostomia, as compared to males. This was also true for the factor of age (OR = 1.105, 95% CI = 1.010–1.209).

## Discussion

Numerous studies have reported the causes, methodology for diagnosing, and impact on the health of dry mouth [1,7,8,10,19]. Therefore, one of the purposes of this study was to clarify the factors associated with the discordance between xerostomia and decreased salivary flow, given that few studies have previously focused on this phenomenon.

Age, sex [4,6,10,20], and medication in particular [21,22] have been shown to affect the dry mouth. However, our present study suggests that these factors may not be associated with the discordance between xerostomia and decreased salivary flow but rather that other unknown factors may be at play. The absence of subjective oral dryness in the group with reduced salivary flow rate could be explained by a denial of symptoms, not only subjective oral dryness [23], but also affective symptoms such as stress and anxiety [24]. In future research, it may be useful to explore these psychological factors, given that the clarification of the factors of discordance and the characteristics of these people may lead to their early identification. These findings may provide relief to people with "latent xerostomia" who have low salivary flow and do not have subjective symptoms.

This study also confirms previous findings [4,10,20] that the decline in USRF, SSRF, and xerostomia tend to be more common in females than males. In addition, dry mouth is more common in the elderly population than in any other age group [1,8,20]. On the other hand, others have reported that aging does not contribute to a decrease in salivary flow [8,25] but rather that salivary disorders in the aging population are usually caused by systemic diseases and their pharmacological treatment [25]. As this study is a cross-sectional study, the results of this study do not accurately demonstrate the effects of aging. Therefore, the results of this study cannot clearly show the effect of aging on dry mouth. Numerous medical conditions, such as Sjögren's syndrome, diabetes, Alzheimer's disease, dehydration, head and neck irradiation, and chemotherapy, can cause or contribute to salivary gland diseases [25]. Such diseases lead to the use of medications, and aging increases the likelihood of developing a variety of diseases. Therefore, it is difficult to independently understand the impact of aging, disease, and medication on the clinical aspects of dry mouth.

The prevalence of low USFR within the general population varies extensively (11.5–47%) [6,7]. In this study, 38.1% of the participants were in the low USFR group, a relatively high percentage compared to previous reports; this may be due to the measurement method, i.e., the visual inspection by a dentist. This method evaluated the correlation with the results of tongue dorsum moisture, while most of the previous studies of USFR were conducted based on the measurement of salivary flow rate. Therefore, comparing the different percentages of low USFR with other reports requires consideration of the methods used for the measurement.

Low SSFR has been reported in the range of 14.8–43% [7]. However, the percentage of patients with low SSFR in our study was 66.0%, higher than in previous reports. Unfortunately, generalizing the prevalence of dry mouth is a difficult task because of the differences in measurement methods, case definitions, and sample characteristics between various studies [1]. The Saxon test has been previously used to measure SSFR, with the salivary flow cutoff value of 1.09 g/2 min [9,26]. We used a cutoff value of 2 g/2 min, and the criteria for this study were stricter than those of the two previous studies, potentially explaining why the percentage of low SSFR was high. However, the cut-off value of the original method of the Saxon test is 2.57 g/2 min [18], but in Japan, 2 g/2 min is used as the standard for the elderly.

Although it is difficult to make a strict comparison due to the different evaluation criteria, we compared the percentage of those who had symptoms of xerostomia and objective reduced salivary flow to the findings of previous studies. In several studies, participants who did not have symptoms of xerostomia but had a low salivary flow rate were found as follows: 16.4% of participants in a survey of the elderly aged 65 and over in South Australia [5]; 10.8% of 50 year-olds enrolled in the Public Dental Health Service in the north of Sweden [23]; and 6.7% of community-dwelling elderly between the ages of 65–84 in Japan [6]. On the other hand, individuals without xerostomia and with an SSFR decline were reported: 10.8% of individuals aged 65–84 who resided in the US Maryland metropolitan area where they were recruited for a study of the Salisbury Eye Evaluation project [26], and approximately 16% of community-dwelling elderly between the ages of 60–81 in Japan [9].

By comparing the percentage of participants who had a low salivary flow rate among those who did not have xerostomia, we concluded that at least 10% (10.3–63.2%) [5,6,9,23,26] might have latent xerostomia. Therefore, based on the findings of our study, we recommend the inclusion of USFR and SSFR tests as oral health assessments for the elderly.

The weaknesses of our study included the small sample size, which can lead to Type II errors. However, we obtained similar results as previous studies due to sex differences, and we maintained minimum statistical power. Additionally, this study targeted those who visited the dental clinic for medical examinations, opening the possibility for selection bias. However, the ratio of males to females and the number of participants within the target age group of this study were similar to the prefecture in which the survey was conducted. This study did not examine exposure and behavioral factors that can affect hyposalivation or xerostomia, such as denture conditions, oral hygiene conditions, smoking, history of radiotherapy or chemotherapy, and level of anxiety or stress. Moreover, salivary flow rate is known to show circadian rhythm and is affected by the sampling time of the day [27]. In the data collection used in this study, measurements were not taken at fixed times of the day and salivary flow rate might have been affected by circadian rhythms. Inconsistency at the time of saliva collection may also be one of the factors making the results unclear. Furthermore, the Saxon test measurement method does not have very high reliability or validity. Using measurement methods with higher reliability for the stimulated salivary flow rate would provide more precise results. Although simple, in this study an appropriate method was adopted for the clinician performing the measurements as the method to measure USFR and SSFR.

## Conclusion

In conclusion, this study shows that of the community-dwelling elderly without xerostomia, 20% have low USFR, and 40% have low SSFR. This study indicates that age, sex, and the number of medications may not be factors in the discrepancy between xerostomia and reduced salivary flow.

## Supporting information

**S1 Dataset. All data of underlying the findings.**
(XLSX)

## Author Contributions

**Conceptualization:** Ichizo Morita, Taketsugu Nomura, Iwane Sugiura, Yuka Kondo, Chisato Kameyama, Kanae Kondo.

**Data curation:** Yoshikazu Abe, Taketsugu Nomura, Seiji Nakashima, Iwane Sugiura, Yujo Inagawa.

**Formal analysis:** Ichizo Morita.

**Funding acquisition:** Yoshikazu Abe, Taketsugu Nomura, Seiji Nakashima, Iwane Sugiura, Yujo Inagawa.

**Methodology:** Hisayoshi Morioka, Yuka Kondo, Chisato Kameyama.

**Writing – original draft:** Ichizo Morita.

**Writing – review & editing:** Ichizo Morita, Hisayoshi Morioka, Yuka Kondo, Chisato Kameyama, Kanae Kondo, Naoji Kobayashi.

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
