## [Decision Letter · Decision Letter 0]

8 Dec 2022

PONE-D-22-29947Discordance between hyposalivation and xerostomia among community-dwelling older adults in JapanPLOS ONE

Dear Dr. Morita,

Thank you for submitting your manuscript to PLOS ONE. After careful consideration, we feel that it has merit but does not fully meet PLOS ONE’s publication criteria as it currently stands. Therefore, we invite you to submit a revised version of the manuscript that addresses the points raised during the review process.

Please submit your revised manuscript by Jan 22 2023 11:59PM. If you will need more time than this to complete your revisions, please reply to this message or contact the journal office at plosone@plos.org. Please include the following items when submitting your revised manuscript:A rebuttal letter that responds to each point raised by the academic editor and reviewer(s). You should upload this letter as a separate file labeled 'Response to Reviewers'.A marked-up copy of your manuscript that highlights changes made to the original version. You should upload this as a separate file labeled 'Revised Manuscript with Track Changes'.An unmarked version of your revised paper without tracked changes. You should upload this as a separate file labeled 'Manuscript'.If applicable, we recommend that you deposit your laboratory protocols in protocols.io to enhance the reproducibility of your results. Protocols.io assigns your protocol its own identifier (DOI) so that it can be cited independently in the future. For instructions see: https://journals.plos.org/plosone/s/submission-guidelines#loc-laboratory-protocols. Additionally, PLOS ONE offers an option for publishing peer-reviewed Lab Protocol articles, which describe protocols hosted on protocols.io. Read more information on sharing protocols at https://plos.org/protocols?utm_medium=editorial-email&utm_source=authorletters&utm_campaign=protocols.

We look forward to receiving your revised manuscript.

Kind regards,

Hadi Ghasemi

Academic Editor

PLOS ONE

Journal Requirements:

"The data for this study were obtained from a survey by the Japanese Ministry of Health, Labor, and Welfare's subvention of the medical system for the elderly. This work was supported by the Japanese Red Cross Toyota College of Nursing Fund for Research Support, received by Professor I Morita."

Reviewers' comments:

Reviewer's Responses to Questions

**Comments to the Author**

1. Is the manuscript technically sound, and do the data support the conclusions?

Reviewer #1: Partly

Reviewer #2: Yes

2. Has the statistical analysis been performed appropriately and rigorously? 

Reviewer #1: I Don't Know

Reviewer #2: Yes

3. Have the authors made all data underlying the findings in their manuscript fully available?

Reviewer #1: Yes

Reviewer #2: Yes

4. Is the manuscript presented in an intelligible fashion and written in standard English?

Reviewer #1: Yes

Reviewer #2: Yes

5. Review Comments to the Author

Reviewer #1: in my opinion, this study was done well ,so it was better to use more exact method for assessment of xerostomia.

How to use saxon test?

it was not clear about sample size?

exclusion criteria or inclusion criteria ?

Reviewer #2: Overall, a very well-done study; several comments/edits:

1. I'm concerned about the question: "Do you worry about dry mouth"; perhaps "Are you experiencing dry mouth" might have led to more accurate answers

2. I would change the wording on a trend toward age as a factor here as the OR is essentially at the null, and for a cross-sectional study, I think one should say age is NOT a factor in the abstract/results sections.

6. PLOS authors have the option to publish the peer review history of their article (what does this mean?). If published, this will include your full peer review and any attached files.

Reviewer #1: No

Reviewer #2: **Yes: **Elliot Abt

---

## [Author Response · Author response to Decision Letter 0]

14 Dec 2022

Response to editors

The “Statement of Ethics,” “Conflicts of Interest,” “Funding Sources,” and “Author Contribution” statements were deleted from the manuscript file. Furthermore, supporting information text was added. The name of the supporting information files was changed from “Dry Mouth Morita dataset.xlsx” to “S1 Dataset.xlsx”. In addition, a “.” was added after the number for the references list.

The “Funding Information” section was updated to include the grant number.

The Japanese Red Cross Toyota College of Nursing distributes funding for research on an annual basis without grant numbers to full-time researchers. As a grant number is unavailable for this funding source, please state whether it should be removed from these sections.

The “Financial Disclosure” section previously stated the following:

“The data for this study were obtained from a survey conducted by the Japanese Ministry of Health, Labor, and Welfare's subvention of the medical system for the elderly. This work was supported by the Japanese Red Cross Toyota College of Nursing Fund for Research Support, and the funding was received by Professor I Morita.”

 Unfortunately, we erroneously listed the source of origin for the data for this study. Herein denotes the corrected sentence. As you pointed out the funders did not play a role in the study design, data collection, analysis, publication decision manuscript preparation.

“The data for this study were obtained from a survey conducted by the Public Interest Incorporated Foundation 8020 Promotion Foundation Dental Health Activity Project in 2018 (No.9, YA). This work was supported by the Japanese Red Cross Toyota College of Nursing Fund for Research Support, and the funding was received by I M. The funders had no role in study design, data collection and analysis, decision to publish, or preparation of the manuscript.”

Please use this information to update the “Financial Disclosure” section appropriately.

4. In your Data Availability statement, you have not specified where the minimal data set underlying the results described in your manuscript can be found. PLOS defines a study's minimal data set as the underlying data used to reach the conclusions drawn in the manuscript and any additional data required to replicate the reported study findings in their entirety.

The following sentence was added to the "Data Availability Statement" section:

 “All data are contained within the manuscript and/or Supporting Information files.” 

5. Please include captions for your Supporting Information files at the end of your manuscript and update any in-text citations to match accordingly.

Supporting information was added to the manuscript. P11L355

We were unable to identify retracted papers in our reference list. If we have erred in this, please provide the reference number for the retracted article. 

Even so, we identified an erratum in reference No. 22 (Turner MD, Ship JA. Dry mouth and its effects on the oral health of the elderly J Am Dent Assoc. 2007). This erratum adds a conflict of interest. 

https://www.sciencedirect.com/science/article/pii/S0002817714614318

CORRECTION

Recent supplements to The Journal of the American Dental Association did not contain disclosure notices regarding conflicts of interest held by the authors. Those notices appear below.

AUTHOR 

 Michael D. Turner, DDS, MD

 Jonathan A. Ship, DMD, FDS RCS (Edin) 

ARTICLE (PAGE NOS.) 

Dry Mouth and Its Effects on the Oral Health of Elderly People (15S-20S) 

DISCLOSURE REPORTED

None reported

None reported

We failed to identify an issue with the description provided. Therefore, we request permission to continue using this reference for our study.

In addition, we have included the following sentences to reference No 22. 

“This article includes a conflict of interest erratum. https://www.sciencedirect.com/science/article/pii/S0002817714614318”

We deleted the publication month from the references list.

 

Response to reviewer 1

in my opinion, this study was done well ,so it was better to use more exact method for assessment of xerostomia.

How to use saxon test?

Thank you for this suggestion. As you highlighted, the Saxon Test does not have high reliability or validity.

As a consequence, we added the following sentence to the discussion.

Revised, P9L251: Furthermore, the Saxon test measurement method does not have very high reliability or validity. Using measurement methods with higher reliability for the stimulated salivary flow rate would provide more precise results.

it was not clear about sample size?

We concur that obtaining a sufficient sample size for this study is important. However, because the study was a secondary analysis, sample size calculation was not performed.

exclusion criteria or inclusion criteria ?

The inclusion criteria for the studies from which data were collected included: community-dwelling, independent, age 70 years or older and informed consent to participate in the study.

The following sentence was added to the study participants section:

Revised, P3L90: The inclusion criteria were as follows: community-dwelling, independent, age 70 years or older and provided informed consent to participate in the study.

 

Response to reviewer 2

Overall, a very well-done study; several comments/edits:

1. I'm concerned about the question: "Do you worry about dry mouth"; perhaps "Are you experiencing dry mouth" might have led to more accurate answers

Thank you for this suggestion. Since the original question is in Japanese, its precise translation in English is difficult. Even so, it is quite possible that some participants interpreted the question to mean “Are you experiencing dry mouth?” as you suggested. However, “Do you worry about having a dry mouth?” is a closer translation of the original question. We will keep your comments in mind for future research.

We changed the sentence of question.

Old: Do you worry about dry mouth?

Revised, P3L96: Do you worry about having a dry mouth?

Revised, P6Table 2: Do you worry about having a dry mouth?

2. I would change the wording on a trend toward age as a factor here as the OR is essentially at the null, and for a cross-sectional study, I think one should say age is NOT a factor in the abstract/results sections.

Thank you for this suggestion. As you highlighted, this was a cross-sectional study. 

We have added the following sentences to the discussion.

Revised, P8L207: As this is a cross-sectional study, the results do not accurately demonstrate the effects of aging. Therefore, the results of this study were unable to provide a clear demonstration of the effect of aging on the dryness of the mouth.

The following sentences were deleted from the discussion.

“This study consolidates that aging may increase the rate of dry mouth, even in the elderly population.”

We have revised the manuscript to replace the word "aging" with "factor of age” in the results section. The changes made to the text are as follows: 

Old manuscript: “Aging was significantly associated (OR = 1.105, 95% CI = 1.010 - 1.209) with low SSFR and xerostomia.”

Revised, P2L51: “Age was a factor that was also significantly associated (OR = 1.105, 95% CI = 1.010 - 1.209) with low SSFR and xerostomia.”

Old manuscript: This was also true for aging (OR = 1.105, 95% CI = 1.010-1.209).

Revised, P7L178: This was also true for the factor of age (OR = 1.105, 95% CI = 1.010-1.209).

---

## [Decision Letter · Decision Letter 1]

23 Jan 2023

PONE-D-22-29947R1Discordance between hyposalivation and xerostomia among community-dwelling older adults in JapanPLOS ONE

Dear Dr. Ichizo Morita,

Thank you for submitting your manuscript to PLOS ONE. After careful consideration, we feel that it has merit but does not fully meet PLOS ONE’s publication criteria as it currently stands. Therefore, we invite you to submit a revised version of the manuscript that addresses the points raised during the review process.

We look forward to receiving your revised manuscript.

Kind regards,

Hadi Ghasemi

Academic Editor

PLOS ONE

Journal Requirements:

Reviewers' comments:

Reviewer's Responses to Questions

**Comments to the Author**

1. If the authors have adequately addressed your comments raised in a previous round of review and you feel that this manuscript is now acceptable for publication, you may indicate that here to bypass the “Comments to the Author” section, enter your conflict of interest statement in the “Confidential to Editor” section, and submit your "Accept" recommendation.

Reviewer #2: (No Response)

Reviewer #3: (No Response)

Reviewer #4: (No Response)

2. Is the manuscript technically sound, and do the data support the conclusions?

Reviewer #2: Yes

Reviewer #3: Partly

Reviewer #4: Yes

3. Has the statistical analysis been performed appropriately and rigorously? 

Reviewer #2: Yes

Reviewer #3: Yes

Reviewer #4: I Don't Know

4. Have the authors made all data underlying the findings in their manuscript fully available?

Reviewer #2: Yes

Reviewer #3: Yes

Reviewer #4: Yes

5. Is the manuscript presented in an intelligible fashion and written in standard English?

Reviewer #2: Yes

Reviewer #3: Yes

Reviewer #4: Yes

6. Review Comments to the Author

Reviewer #2: (No Response)

Reviewer #3: Interesting research work done by the authors, however, some alternations need to be considered to improve the quality of the manuscript as follows:

1- Despite the correction of the inclusion criteria in the first revision, the exclusion criteria have not been determined yet. Please include your exclusion criteria. (it can be for example: having a hearing or language impairment that could affect the interview or having a physical or mental disability that could interfere with the oral exam or …)

2- Is there any more appropriate method to measure USFR or SSFR?

3- It is better to consider about exposures and behavioral factors that can effect on xerostomia or hyposalivation such as Condition of dentures, if any, and how to use them, oral hygiene condition, Smoking, history of radiotherapy or chemotherapy and the level of anxiety or stress and etc. Considering that the current study is a secondary analysis based on primary data, was there any information about these factors in the primary data?

4- It is also preferable to cite recent sources that were published less than 5 years ago.

Best wishes.

Reviewer #4: 1. The study was well done, but there is a concern and it is the time of saliva sampling. Considering the effect of circadian rhythm on salivary flow, if it is not considered, should be discussed as a limitation of the study.

2. There are some specific questionaries for the diagnosis of xerostomia, please add a reference to your mentioned question that was used as a diagnosis tool.

7. PLOS authors have the option to publish the peer review history of their article (what does this mean?). If published, this will include your full peer review and any attached files.

Reviewer #2: **Yes: **Elliot Abt

Reviewer #3: No

Reviewer #4: **Yes: **Mahdieh-Sadat Moosavi

---

## [Author Response · Author response to Decision Letter 1]

16 Feb 2023

Response to Reviewer 3

1- Despite the correction of the inclusion criteria in the first revision, the exclusion criteria have not been determined yet. Please include your exclusion criteria. (it can be for example: having a hearing or language impairment that could affect the interview or having a physical or mental disability that could interfere with the oral exam or …)

Thank you for your suggestion. We added the following sentence to the Study Participants section:

Revised, P3L80:

The exclusion criteria were as follows: hearing or language impairment that could affect the interview, physical or mental disability that could interfere with the oral exam, symptoms, or need for treatment for dental disease.

2- Is there any more appropriate method to measure USFR or SSFR?

This study used visual inspection of the oral cavity by a dentist to measure the unstimulated salivary flow rate (USFR), and the Saxon test was used to measure the stimulated salivary flow rate (SSFR).

As a method of measuring USFR, in addition to the method used in this study, there is a method of collecting saliva. Saliva discharged into the oral cavity in a resting state is exhaled or aspirated, collected, and weighed. Saliva discharged into the oral cavity in a resting state is collected, that is, voided or suctioned and weighed (Tomson, 2015: Ref. no. 1). These methods have the advantage of being highly objective, but stimulation of saliva collection is added. SSFR can be measured using the sense of taste, that is, stimulation by citric acid solution, in addition to stimulation by mastication. Each method has its advantages and disadvantages. Because saliva measurements were performed by a clinical dentist, a method that could be performed by a clinician was adopted.

We added the following sentence to the Discussion section.

Revised, P10L266: Although simple, in this study an appropriate method was adopted for the clinician performing the measurements as the method to measure USFR and SSFR.

3- It is better to consider about exposures and behavioral factors that can effect on xerostomia or hyposalivation such as Condition of dentures, if any, and how to use them, oral hygiene condition, Smoking, history of radiotherapy or chemotherapy and the level of anxiety or stress and etc. Considering that the current study is a secondary analysis based on primary data, was there any information about these factors in the primary data?

The primary dataset did not include other information such as oral hygiene conditions, smoking, history of radiotherapy or chemotherapy, and level of anxiety or stress. Information on the number of teeth and dentures used was included in the primary dataset. However, this information was considered age-related and was excluded from the analysis plan. In addition, due to the limited amount of data available, we did not include information on the number of teeth and dentures used to limit the number of explanatory variables for the multinomial logistic regression analysis.

We added the following sentence to the Discussion section.

Revised, P9L251: This study did not examine exposure and behavioral factors that can affect hyposalivation or xerostomia, such as denture conditions, oral hygiene conditions, smoking, history of radiotherapy or chemotherapy, and level of anxiety or stress.

4- It is also preferable to cite recent sources that were published less than 5 years ago.

Best wishes.

This comment provided us with an opportunity to look through the current literature again. Recognizing that many new papers on dry mouth have been published since we finished writing this paper, we reconfirmed the importance of research in this field. However, with many broad-based reports, it was difficult to determine which additional information would be desirable to add to our paper. We would appreciate if you could specify the information we should cite in our paper.

 

Response to Reviewer 4

1. The study was well done, but there is a concern and it is the time of saliva sampling. Considering the effect of circadian rhythm on salivary flow, if it is not considered, should be discussed as a limitation of the study.

Saliva flow is known to be affected by the sampling time of the day. However, in the data collection used in this study, it was difficult to measure at the same time of day. Therefore, regarding the concern that saliva sampling was not performed at the same time of day, we added the following to the limitations of the study.

Revised, P9L254: Moreover, salivary flow rate is known to show circadian rhythm and is affected by the sampling time of the day [27]. In the data collection used in this study, measurements were not taken at fixed times of the day and salivary flow rate might have been affected by circadian rhythms. Inconsistency at the time of saliva collection may also be one of the factors making the results unclear.

One reference was added.

27 Dawes C. Circadian rhythms in human salivary flow rate and composition. J Physiol. 1972;220(3):529-245.

2. There are some specific questionnaires for the diagnosis of xerostomia, please add a reference to your mentioned question that was used as a diagnosis tool.

The question in this study used to ask about subjective dry mouth is recommended by the Japanese Ministry of Health, Labor, and Welfare as one of the questionnaire items to evaluate oral function. This question appears to belong to the lineage of questions in Tomson et al. (2007, 2011) and Ohara et al. (2016). These studies were added to the references for the subjective question of dry mouth.

Revised, P3L87: The presence of xerostomia was assessed with one question: "Do you worry about having a dry mouth?" [14-16]. Answer options were either yes or no.

The following references were added.

14 Thomson WM. Measuring change in dry-mouth symptoms over time using the Xerostomia Inventory. Gerodontology. 2007;24(1):30-35.

15 Thomson WM, van der Putten der, de Baat Baat, Ikebe K, Matsuda K, Enoki K, et al. Shortening the xerostomia inventory. Oral Surg Oral Med Oral Pathol Oral Radiol Endod. 2011;112(3):322-327.

16 Ohara Y, Hirano H, Yoshida H, Obuchi S, Ihara K, Fujiwara Y, et al. Prevalence and factors associated with xerostomia and hyposalivation among community-dwelling older people in Japan. Gerodontology. 2016;33(1):20-27.

---

## [Decision Letter · Decision Letter 2]

22 Feb 2023

Discordance between hyposalivation and xerostomia among community-dwelling older adults in Japan

PONE-D-22-29947R2

Dear Dr. Ichizo Morita,

We’re pleased to inform you that your manuscript has been judged scientifically suitable for publication and will be formally accepted for publication once it meets all outstanding technical requirements.

Kind regards,

Hadi Ghasemi

Academic Editor

PLOS ONE

Additional Editor Comments (optional):

Reviewers' comments:

Reviewer's Responses to Questions

**Comments to the Author**

1. If the authors have adequately addressed your comments raised in a previous round of review and you feel that this manuscript is now acceptable for publication, you may indicate that here to bypass the “Comments to the Author” section, enter your conflict of interest statement in the “Confidential to Editor” section, and submit your "Accept" recommendation.

Reviewer #4: All comments have been addressed

2. Is the manuscript technically sound, and do the data support the conclusions?

Reviewer #4: Yes

3. Has the statistical analysis been performed appropriately and rigorously? 

Reviewer #4: I Don't Know

4. Have the authors made all data underlying the findings in their manuscript fully available?

Reviewer #4: Yes

5. Is the manuscript presented in an intelligible fashion and written in standard English?

Reviewer #4: Yes

6. Review Comments to the Author

Reviewer #4: All comments have been addressed.By adding these references, it becomes possible to resolve the limitations for future researches.

7. PLOS authors have the option to publish the peer review history of their article (what does this mean?). If published, this will include your full peer review and any attached files.

Reviewer #4: **Yes: **Mahdieh-Sadat Moosavi

---

## [Editor Report · Acceptance letter]

24 Feb 2023

PONE-D-22-29947R2 

Discordance between hyposalivation and xerostomia among community-dwelling older adults in Japan 

Dear Dr. Morita:

I'm pleased to inform you that your manuscript has been deemed suitable for publication in PLOS ONE. Congratulations! Your manuscript is now with our production department. 

Kind regards, 

on behalf of

Dr. Hadi Ghasemi 

Academic Editor

PLOS ONE